# Overcoming Resistance to Immune Checkpoint Inhibitor Therapy Using Calreticulin-Inducing Nanoparticle

**DOI:** 10.3390/pharmaceutics15061693

**Published:** 2023-06-09

**Authors:** Sri Vidhya Chandrasekar, Akansha Singh, Ashish Ranjan

**Affiliations:** Department of Physiological Sciences, College of Veterinary, Oklahoma State University, Stillwater, OK 74078, USA; srchand@okstate.edu (S.V.C.);

**Keywords:** immunogenic cell death, calreticulin, nanoparticle, colon carcinoma, immune checkpoint inhibitor, immunoresistance

## Abstract

Nanoparticles (NPs) have the ability to transform poorly immunogenic tumors into activated ‘hot’ targets. In this study, we investigated the potential of a liposome-based nanoparticle (CRT-NP) expressing calreticulin as an in-situ vaccine to restore sensitivity to anti-CTLA4 immune checkpoint inhibitor (ICI) in CT26 colon tumors. We found that a CRT-NP with a hydrodynamic diameter of approximately 300 nm and a zeta potential of approximately +20 mV induced immunogenic cell death (ICD) in CT-26 cells in a dose-dependent manner. In the mouse model of CT26 xenograft tumors, both CRT-NP and ICI monotherapy caused moderate reductions in tumor growth compared to the untreated control group. However, the combination therapy of CRT-NP and anti-CTLA4 ICI resulted in remarkable suppression of tumor growth rates (>70%) compared to untreated mice. This combination therapy also reshaped the tumor microenvironment (TME), achieving the increased infiltration of antigen-presenting cells (APCs) such as dendritic cells and M1 macrophages, as well as an abundance of T cells expressing granzyme B and a reduction in the population of CD4+ Foxp3 regulatory cells. Our findings indicate that CRT-NPs can effectively reverse immune resistance to anti-CTLA4 ICI therapy in mice, thereby improving the immunotherapeutic outcome in the mouse model.

## 1. Introduction

The immunotherapy of solid tumors using immune checkpoint inhibitors (ICI) that block checkpoint molecules (e.g., PDL-1, CTLA-4, etc.) to improve effector T cell activity has significantly expanded in recent years [1]. Effective antitumor immunity should facilitate the recognition, uptake, and cross-presentation of released antigens by antigen-presenting cells (APCs) to the T cells, as well as the tumor-specific infiltration of cytotoxic T cells for sustained tumor-killing until clearance [2,3,4]. Most ICIs and immunomodulators, when employed as a monotherapy, stimulate a small number of the essential stages outlined above, thereby presenting significant hurdles for their translation to clinical use [5,6]. One approach to overcome this barrier can be to target multiple immune pathways to achieve the needed desired response. More than 50 registered immuno-oncology trials are focusing on combination immunotherapies to leverage the synergistic benefits of monotherapies for improved survival rates [5,6,7,8].

The value of combination therapy can be especially advantageous in cases where physiological factors reduce response to ICIs. For example, CTLA-4, a cytotoxic T-lymphocyte-associated antigen expressed on T cells, inhibits its activation following interaction with tumor-resident immunosuppressive myeloid populations. Anti-CTLA therapy works by blocking the CTLA-4 inhibitory signal, which can help activate T cells and enhance their ability to recognize and attack cancer cells. Several anti-CTLA-4 therapies have been developed and approved for the treatment of various types of cancer, and these agents can slow down tumor growth clinically [9]. While anti-CTLA4 monotherapy can induce some degree of antitumor activity, the presence of Tregs, low mutation burden, and high expression of PD-L1 in the tumor microenvironment can limit its effectiveness [10,11]. Herein, we aimed to understand the role of in situ vaccination (ISV)-based immunotherapy using our calreticulin nanoparticle (CRT-NP) and its correlation with anti-CTLA-4 therapeutic outcomes in mice bearing Colorectal cancer (CRC).

Colorectal cancer (CRC) is known to manifest a variety of immunosuppressive mechanisms to evade antitumor immunity. CRT is a calcium-binding protein that is found in the endoplasmic reticulum of cells. CRT upregulation in tumors improves the tumor infiltration of leukocytes to aid cancer immunotherapy with ICIs (e.g., anti-CTLA-4). We have shown that our novel calreticulin nanoparticle (CRT-NP) upregulates the CRT damage-associated molecular patterns (DAMPs) in solid tumors [12]. DAMP is a marker of immunogenic cell death (ICD), which is a form of programmed cell death that can induce an immune response against cancer cells. Although promising, whether CRT upregulation in tumors supports an effective antitumor immune response with anti-CTLA-4 is currently unknown. Therefore, the effects of anti-CTLA-4 ICI and CRT-NPs on CRC tumor progression were analyzed as monotherapies, respectively, as well as in combination therapy to enhance the translatability of ICI use in patients, and potentially reduce their dose-dependent toxicities. Data from this study suggest that overcoming ICI immunoresistance using CRT-NPs could be a promising innovative strategy for cancer treatment by unleashing the unique potential of the immune system.

## 2. Materials and Methods

### 2.1. Materials

DOTAP (1,2-dioleoyl-3-trimethylammonium-propane) and cholesterol lipoids were purchased from Avanti Polar Lipids, Inc., Alabaster, AL, USA. Nanoplasmids (NTC 9385R-MCS) containing the CRT gene and Green Fluorescent Protein (GFP) were purchased from Aldevron (Lincoln, NE, USA). Chloroform (C2432), 2-Propanol (I9516), and ethyl alcohol (E7023) were purchased from Sigma-Aldrich, St. Louis, MO, USA. HEPES, sterile 1M buffer (J848), and Triton^®^ X-100 (0694) were purchased from VWR life science, Radnor, PA, USA. Agarose Precast Gel (3% TBE EtBr Wide Mini Ready—20-well) (1613030) was purchased from Bio-Rad, Hercules, CA, USA. DMEM (11965092), fetal bovine serum, FBS (10082147), penicillin–streptomycin—Pen Strep (15140122), PBS (10010023), and collagenase IV (17104019) were purchased from Thermo Fisher/Gibco, Waltham, MA, USA. Anti-mouse CTLA-4 antibodies (BE0034) were purchased from BioXCell, West Lebanon, NH, USA. The following fluorochrome-conjugated monoclonal antibodies (mAbs) for flow cytometry were purchased from BioLegend, San Diego, CA, USA: APC-Cy7 anti-CD45 (103115), PerCP anti-CD3 epsilon (100325), PE-Cy5 anti-CD4 (100410), PE-Cy7 anti-CD8a (100721), PE anti-Granzyme B (372208), BV510 anti-INFγ (505841), BV605 anti-PD-1 (135219), BV510 anti-Ly-6G (127633), PE anti-Ly-6C (128007), BV650 anti-CD11b (101239), APC anti-CD206 (141707), BV785 anti-MHC II (107645), BV421 anti-CD11c (117329), AF488 anti-CD86(105017), and Apotracker™ Green (427403). Lipofectamine™ 2000 (LF™2000) (11668030), SYTOX™ Blue Dead Cell Stain (S34857), TRIzol RNA isolation reagent (15596026), and 10X RBC Lysis Buffer (00-4300-54) were purchased from Invitrogen, Waltham, MA, USA. MTT reagent (T-030-1) was purchased from Gold Biotechnology, Olivette, MO, USA. Doxorubicin hydrochloride (DOX) (J64000.MF) was purchased from ThermoScientific, Waltham, MA, USA. CT26 murine colon cancer cell line (CRL-2638) was purchased from the American Type Culture Collection (ATCC), Manassas, VA, USA. Liberase (5401119001) was procured from Life Technologies, NY, USA. Transcription factor buffer set (562574) was purchased from BD Biosciences, San Jose, CA, USA. iScript™ gDNA Clear cDNA Synthesis Kit (1725034) and iTaq Universal SYBR Green Supermix (1725124) were purchased from Bio-Rad, Hercules, CA, USA. Murine Calreticulin (CRT) and GAPDH primers were designed by and purchased from Integrated DNA Technologies, Inc., Coralville, IA, USA.

### 2.2. Preparation and Characterization of CRT-NP

For CRT-NP synthesis, cationic liposomes were first generated using the thin-film hydration method. Briefly, DOTAP and cholesterol in equimolar ratio were co-dissolved in chloroform and dried under vacuum using a rotatory evaporator. The dried lipid film was rehydrated with 10 mM HEPES buffer and sonicated for 10 min. The resulting liposomes were then extruded 5 times through two stacked polycarbonate nanopore filters (200 nm pore diameter) to obtain a homogenous suspension and they were stored at 4 °C until use. For CRT-plasmid loading, liposomes were diluted and gently mixed with the full-length cloned DNA of murine CRT at a 10:1 ratio (lipid/pDNA), yielding a total volume of 100 µL DNA/lipoplex solution. The CRT gene was cloned into an antimicrobial resistance (AMR) gene-free NTC plasmid backbone (Nature Technology Corp., Lincoln, NE, USA), an ideal non-viral vector for translation to human trials, vs. pCMV3 expression systems that we published previously with AMR genes [1]. The suspension was incubated at room temperature for 30 min to ensure CRT-NP formation. The hydrodynamic diameter, surface charge, and polydispersity index (PDI) of liposomes before and after encapsulation of pCRT were analyzed using dynamic light scattering (DLS) and zeta potential measurements with a Brookhaven ZetaPALS instrument (Holtsville, NY, USA) for 3 days to determine their stability. Additionally, transmission electron microscopy (TEM) (JEOL JEM-2100 USA, Peabody, MA, USA) of synthesized CRT-NPs was similarly conducted to evaluate the morphology and stability for 3 days. Agarose gel electrophoresis was used to determine the encapsulation efficiency of CRT-NPs by centrifuging the liposomes at 13,000× *g* and collecting the supernatant and pellet separately. Next, 10% 100X triton was added to the pellet to lyse CRT-NPs. The supernatant and different concentrations of free positive pCRT control (1.25 µg, 2.5 µg, and 5 µg) were run on agarose gel at 100 V. Free plasmid migration was observed at ~60 min.

### 2.3. Evaluation of CRT-NP Cytotoxicity, Transfection, Gene Expression, and ICD against CT26 Cells

For the dose-dependent cytotoxicity assessment, CRT-NPs (0.5, 1, 1.5, 2, 2.5, and 3 µg/well) were added to 2 × 10^3^ CT26 cells in a serum-free medium and incubated for 5–6 h at 37 °C. Then, the medium was replaced with complete DMEM medium, and cells were incubated for 48 h at 37 °C at 5% CO_2_. MTT reagent was added (10 µL = 0.05 mg/well) and the plates were read using a spectrophotometer at 540 nm to determine the optical density (OD). To determine the transfection efficiency, 2 *×* 10^5^ CT26 cells/well seeded in a 12-well plate were transfected with control EGFP-NPs (eukaryotic GFP plasmid + liposomes) for 5–6 h and incubated for 48 h at 37 °C at 5% CO_2_. Fluorescent imaging was performed with the GFP filter cube (ex/em of 469/525 nm) using a Biotek Cytation 5-cell imaging multimode reader (Winooski, VT, USA), and the images were acquired using Gen5 Image + software version 3.08.01. 2 *×* 10^5^ CT26 cells/well in the 12-well plate were also transfected with EGFP-NP, stored at 4 °C on various days (day 0, 1, 2, and 3), and fluorescent images were obtained at 14, 24, and 48 h to confirm the plasmid expression and stability of the nanoparticles. Finally, to verify the gene expression of CRT-NPs in the transfected cells, untreated control cells, pCRT, and blank liposome-treated cells were incubated for 8 h. RNA was isolated from these cells using TRIzol RNA isolation reagent (Invitrogen™). Two micrograms of RNA was used for cDNA synthesis with the iScript™ clear cDNA Synthesis Kit in a Bio-Rad thermal cycler (Bio-Rad, Hercules, CA, USA). The CRT gene expression was quantified using the QuantStudio 6 Pro real-time PCR system (Applied Biosystems, Thermo Fisher, Waltham, MA, USA), with GAPDH used as an internal control. The primer sequences used for CRT were as follows: FW-5′-AAAGGACCCTGATGC TGCCAAG-3′, RW- 5′-TCAGGGATGTGCTCTGGCTTG T-3′; GAPDH, FW–5′-CATCAC TGCCACCCAGAAGACTG-3′ RW-5′-ATGCCAGTGAGCTTCCCGTTCAG-3′. To establish ICD, 2 *×* 10^5^ CT26 cells/wells were seeded in a 12-well plate and transfected for 5 h with CRT-NP (2 µg DNA/well), LF^TM^2000 + pCRT (3:1 ratio), and DOX (2.5 µM/well) as a positive control for ICD. After 48 h of incubation, cells were scraped and stained with Apotracker™ Green and SYTOX™ Blue Dead Cell Stain and analyzed in a NovoCyte flow cytometer (Agilent, Santa Clara, CA, USA).

### 2.4. In Vivo Study Design

All animal-related procedures were approved and carried out under the guidelines of the Oklahoma State University Animal Care and Use Committee. CT26 cells cultured in complete medium were harvested at ~80% confluency. Then, 1 *×* 10^6^ CT26 cells re-suspended in 100 µL of PBS were subcutaneously injected in the left flank region of 10–12-week-old male BALB/c mice. Seven days post-inoculation, n = 6 mice/group were randomized as follows: (1) Control, (2) CRT-NP, (3) ICI, and (4) CRT-NP+ICI. Three CRT-NP intratumoral injections (20 µg DNA/injection) were administered 3 days apart, and anti-CTLA-4 ICI (200 µg/dose) was administered 72 h after each CRT-NP injection intraperitoneally (Figure 2A). Mice tumor volumes were measured daily by serial caliper (General Tools Fraction™, New York, NY, USA) measurements using the formula (length *×* width^2^)/2, where the length represented the longest dimension.

### 2.5. Flow Cytometry Assessment of Tumor Immune Microenvironment

Tumors were collected on day 20 post-inoculation, and were physically fragmented, digested with 100 U/mL of liberase solution, filtered through a 70 µm cell strainer (Corning Inc., Corning, NY, USA), and incubated in RBC lysis buffer for 10 min. Cells were resuspended in flow buffer (PBS + 2% BSA) and stained on ice for 30 min in the dark with above mentioned fluorochrome-conjugated antibodies using a separate panel for T cells and myeloid cells. Intracellular antigen staining of IFNγ, Granzyme-B, and CD206 was performed by fixing and permeabilizing cells with a transcription factor buffer set (BD Biosciences, San Jose, CA, USA) for 20 min on ice and incubating cells with specific antibodies upon washing. All acquisitions were performed on an LSR II flow cytometer (BD Biosciences) within 24 h. Compensations were performed with unstained and single-stained cells. FlowJo software v.10.2 (Treestar Inc., Ashland, OR, USA) was used for data analysis and cells were gated as follows: CD45+ (total leukocytes), CD45+ CD3+ (total T cells), CD45+ CD3+ CD4+ CD8− (TH, CD4+ T helper cells), CD45+ CD3+ CD8− CD4+ INFγ (activated CD4 T cells), CD45+ CD3+ CD4− CD8+ (TC, CD8+ T cells), CD45+ CD3+ CD4− CD8+ GZMB+ & INFγ (effector cytotoxic T cells), CD45+ CD3+ CD4− CD8+ PD-1+ (exhausted CD8+ T cells) CD45+ CD11c+ (DC, dendritic cells), CD45+ CD11c+ MHC-II+ CD86+ (activated DCs), CD45+ CD11b+ (macrophages), CD45+ CD11b+ CD86+ MHC-II+ (activated M1 macrophages), CD45+ CD11b+ 206+ (M2 macrophages), CD45+ CD11b+ Ly6C^Hi^ Ly6G− (monocytic myeloid-derived suppressor cells, M-MDSC) & CD45+ CD11b+ Ly6C^Lo^ Ly6G+ (polymorphonuclear-MDSC, PMN-MDSC).

### 2.6. Statistical Analysis

Statistical analyses were performed using GraphPad Prism 6.0 software (GraphPad Software Inc., La Jolla, CA, USA). Data are presented as mean ± SEM unless otherwise indicated. Analysis of differences between 2 normally distributed test groups was performed using an unpaired t-test assuming unequal variance. P values less than 0.05 were considered significant and represented as * *p* < 0.05, ** *p* < 0.005, *** *p* < 0.0005, **** *p* < 0.0001.

## 3. Results

### 3.1. CRT-NP Characterizations and Stability

The hydrodynamic diameter, polydispersity index (PDI), and zeta potential of blank and CRT-NPs were measured using DLS. Blank liposome NPs showed a zeta potential of +49 mV, a mean size of 157 ± 5.27 nm, and a PDI below 0.15. CRT-NPs had larger diameters of 299 ± 7.09 nm with a PDI of 0.228 ± 0.032 and a decreased zeta potential of +20 mV ± 1.89 (Figure 1A). CRT-NPs stored at 4 °C showed no major variations in hydrodynamic diameter, while their zeta potential slightly decreased to +14 mV on day 3. The morphological stability of the CRT-NPs was confirmed using transmission electron microscopy for up to three days. The CRT-NPs exhibited a typical spherical morphology with an average size of approximately 280 nm (Figure 1B). The efficient CRT plasmid encapsulation in the NPs was also confirmed by the agarose gel retardation assay, which showed the absence of plasmid migration from the unlysed CRT-NPs. In contrast, the free plasmid (~2.6 µg) released from CRT-NPs with triton showed a band corresponding to a 2.5 µg free plasmid band (Figure 1C).

### 3.2. Gene Delivery Efficiency, In Vitro Cytotoxicity, Gene Expression, and ICD-Inducing Ability of CRT-NPs

To visualize the transfection efficacy of NPs, CT26 cells were transfected with EGFP-NP lipoplexes (CLs/pDNA) (Figure 1D). Significant uptake and expression of GFP were observed for EGFP-NPs compared to the untreated control, blank liposome, and free EGFP plasmid. The plasmid stability of the NP complex was confirmed by transfecting CT26 cells with EGFP-NPs stored at 4 °C for up to three days. The data showed no difference in the transfection rates of EGFP-NP over 3 days (Appendix A).

The cytotoxicity of CRT-NPs in CT26 cells was assessed using the MTT assay at 48 h post-transfection (Figure 1E). Dose-dependent cytotoxicity was observed, with significantly reduced cell viability at higher concentrations of CRT-NPs. The enhanced cytotoxicity of the CRT-NPs was associated with a significant increase in CRT gene expression in CT26 cells (Figure 1F). From the apoptotic assay, CRT-NPs showed an ICD comparable to positive controls such as DOX-treated and transfected LF™2000 + pCRT cells, indicating an immunogenic phenotype (Figure 1G).

### 3.3. CRT-NP Combined Anti-CTLA-4 Repressed CT26 Colon Tumor Progression In Vivo

Mice showed tumor remission with CRT-NP (~30% vs. control) and anti-CTLA-4 (~20% vs. control) treatment (Figure 2B). In contrast, the combination of CRT-NPs and antiCTLA4 improved efficacy, inducing tumor reductions of almost 70% relative to the untreated control. These were also confirmed by the tumor weight analysis that corresponded with the tumor volume of the sacrifice (Figure 2C).

### 3.4. Myeloid Cell Dynamics with CRT-NPs, antiCTLA4I, and CRT-NPs + antiCTLA4

CRT-NPs + antiCTLA4 induced superior CD11c+ dendritic cell infiltration compared to monotherapies and untreated controls (Figure 3A). The frequencies of activated DCs (CD11c+ MHC-II+ CD86+) (Figure 3B,C) and M1 macrophages (CD11b+ MHC-II+ CD86+) (Figure 3D) increased ~5-fold compared to respective controls. The expression of the M2 macrophage marker, CD206, also significantly reduced in CD11b+ macrophages in CRT-NP+ICI-treated tumors compared to all other groups (Figure 3E), whereas the M-MDSC (CD11b+ LY6C^Hi^ LY6G-) population decreased ~4-folds vs. the CRT-NP and control group and ~2-fold vs. the ICI group (Figure 3F).

### 3.5. T Cell Dynamics with CRT-NPs, antCITLA4, and CRT-NP+antiCTLA4

Immune infiltrations detected using the pan-leukocyte marker CD45 showed significantly higher populations in tumors treated with CRT-NP+ICI (Figure 4A). These results were corroborated by the infiltration of CD3+ and CD8+ T cells and enhanced expressions of INFγ on CD4+ T cells (Figure 4B,C) in the combinations relative to the untreated control and monotherapies. CRT-NP+ICI also significantly increased cytotoxic CD8+ GZMB+ T cells compared to monotherapies and reduced exhaustion markers PD1 on CD8 T cells (Figure 4D,E). A higher ratio of CD8+ GZMB+ to CD4+ FOXP3+ was also observed in the CRT-NP+ICI treatment relative to other groups (Figure 4F).

## 4. Discussion

Immune checkpoints play a critical role in inhibiting uncontrolled T cell responses after initial stimulation [13]. CRC tumors are characterized by a microenvironment rich in immunosuppressive cells, such as tumor-associated macrophages (TAMs), myeloid-derived suppressor cells (MDSCs), and TIE-2-expressing monocytes (TEMs), which can contribute to creating an immunosuppressive microenvironment [14]. These immunosuppressive TAMs and MDSCs directly influence the cytotoxic activity of T cells by engaging with checkpoints such as CTLA-4, PD-1, TIM-3, and LAG-3 on T cells [15,16]. ICIs have shown to improve the antitumor response against solid tumors, including CRC, by releasing the breaks in the immune system [13,17,18,19,20,21,22,23,24]. However, ICIs have limited efficacy in patients as monotherapies due to adverse reactions such as immunotherapy-related adverse events (irAEs), innate or acquired resistance, and genomic and tumor heterogeneity [23,25,26]. The ICI-induced response is complex, heterogeneous, and inconsistent, emphasizing the need to better understand its challenges and mitigation strategies [25,27,28,29,30,31]. Herein, we aimed to understand the role of CRT-NPs in influencing ICI activity in the CT26 colon cancer model via in vivo efficacy and tumor-immune characterizations.

Calreticulin (CRT) is an endoplasmic-reticulum-resident chaperone that functions as a DAMP and plays a crucial role in tumor antigen recognition [32,33,34]. The ICD-induced expression of CRT on cancer surfaces can attract phagocytes, such as DCs and macrophages, which enhances antigen presentation [35,36,37]. We evaluated the ability of CRT-NPs to induce ICD in CT26. CRT-NPs based on the cationic liposome platform can interact with the negatively charged cell membrane, leading to the formation of liposome–cell complexes [12]. We employed DOTAP and cholesterol lipid-based cationic liposomes because they are biodegradable and can achieve efficient plasmid loading and transfection [38,39]. Our CRT-NPs with a positive zeta were optimal for transfection, like other cationic carriers [40,41]. Our data also showed that the NP diameter increased following plasmid loading, but this did not affect encapsulation efficiency, as shown by the agarose gel assay, which demonstrated no plasmid migration (Figure 1A,B). These findings are consistent with the related research based on cationic liposomes (DOTAP and cholesterol) loaded with different concentrations of plasmid DNA [12,42,43]. Cationic liposome complexes can enter the cell through various pathways, including endocytosis, fusion, or direct penetration to release their encapsulated genetic material, ultimately leading to the expression of the encoded protein [44]. To correlate CRT-NP-induced CRT expression with ICD, we chose DOX as a positive control since it is known to generate cell stress and induce CRT translocation on the surface of cancer cells [45]. We found that, at the selected concentration, CRT-NPs induced ICD similarly to DOX, thereby leading us to hypothesize that CRT expressed using CRT-NPs can act as a danger signal to re-engineer the immune system for improved sensitivity to anti-CTLA4.

Based on our clinically relevant in vitro findings, we designed an in-situ vaccination strategy (ISV) to investigate our hypothesis. ISV involves the delivery of an immune-stimulating agent directly into a tumor, which can trigger an antitumor immune response that targets not only the injected tumor but also metastatic lesions and uninjected tumors [46]. We and others have shown the potential of in situ vaccination to improve the effectiveness of cancer treatment [47,48]. ISV has become an attractive strategy for cancer treatment with recent advances in immunotherapy. One of the advantages of ISV is that it allows for the localized delivery of therapeutic agents, minimizing systemic exposure and reducing side effects. For instance, a recent phase I clinical trial showed the safety of an ISV approach using a toll-like receptor 9 (TLR9) agonist in patients with advanced solid tumors [49]. Additionally, it can overcome some of the challenges associated with systemic administration, such as poor bioavailability, rapid clearance, and nonspecific targeting. Phase I and II trials evaluating the efficacy of intratumoral injection of a genetically modified herpes simplex virus (T-VEC) in patients with advanced melanoma, and CD40 agonist antibody (CP-870,893) in patients with pancreatic cancer, have already demonstrated its feasibility against both superficial and deep-seated tumors [50,51]. Thus, we posit that administering CRT plasmids via liposomes (CRT-NP) directly to the tumors would stimulate ICD and effectively augment the real-time antitumor immune response in vivo.

To investigate the potential synergy between CRT-NP ISV and anti-CTLA immunotherapy, we employed a syngeneic colon carcinoma model (CT26) due to its classification as a T-cell-inflamed tumor for an effective ICI outcome [52,53,54]. However, one mechanism by which CT-26 colon tumor cells can evade the immune system and ICI immunotherapy is through the presence of high levels of regulatory T cells (Tregs) in the tumor microenvironment. Research conducted on CT26 models has demonstrated that regulatory T cells (Tregs) with high levels of TIM3+ accumulate in the tumor microenvironment prior to the exhaustion of CD8+ and CD4+ T cells. When a TIM-3 blockade was administered to Tregs that had infiltrated the tumor, it induced the production of IL10, a cytokine that is recognized to contribute to T cell exhaustion. Moreover, the depletion of Tregs and neutralization of TIM-3 in the early stages of tumor growth led to the substantial and long-lasting regression of the tumor [55,56]. Interestingly, our study found that the combinatorial regimen of CRT-NPs synergistically improved the antiCTLA4 effect, inhibiting the CT26 colon tumor growth relative to the monotherapies (Figure 2). To understand the mechanistic significance of this approach, we first characterized the myeloid cells in the tumor microenvironment since ICD induces APC activation. We noted an increased infiltration of activated dendritic cells and M1 macrophages, and a reduction in MDSCs with the combination treatment (Figure 3). MDSCs inhibit the immune-suppressive T cells in CT26 and other tumor types through various mechanisms such as high levels of arginase I (Arg), inducible nitric oxide synthase (iNOS), and others [57]. MDSCs also interact with tumor macrophages and dendritic cells to influence their activity and upregulate the activity of Tregs through their production of IL-10, TGFβ, IFNγ, and CD40–CD40L interactions. Our data showed that CRT-NPs in combination with anti-CTLA4 induced a significant reduction in the PMN-MDSC to M-MDCS ratio. Tumor-resident M-MDSCs suppress T cell activation and rapidly differentiate into M2 macrophages, which are features typically associated with poor prognosis in various solid tumor types [57]. These MDSCs are also implicated in resistance against ICI. Therefore, the ability of CRT-NPs to reprogram the tumor microenvironment via MDSCs likely improved both the infiltration and functional capacity of antigen-presenting cells [57], which then correlated with the overall increase in CD8+ T cell infiltration and anti-CTLA4 synergy, improving the infiltration of granzyme-positive CD8+ and CD4+ T cells and reducing the CD4+ Foxp3 cell population [10,58]. These results lead us to propose that CRT-NPs reengineer the TME by reducing the population of immunosuppressive myeloid cells such as MDSCs, which, in turn, augments the anti-CTLA4 effect and the resultant T cell action, enhancing the immune cell cytotoxicity and tumor regression.

## 5. Conclusions

In summary, combining CRT-NPs with anti-CTLA-4 immunotherapy could be a promising approach to improve the effectiveness of immunotherapy in treating certain types of cancer, specifically, those with T-cell-inflamed tumors such as the CT26 colon carcinoma model used in this study. This combination could potentially help overcome the immunosuppressive nature of the tumor microenvironment and enhance the immune response against cancer cells, leading to better treatment outcomes for patients who may not respond well to standard immunotherapy.

## Figures and Tables

**Figure 1 pharmaceutics-15-01693-f001:**
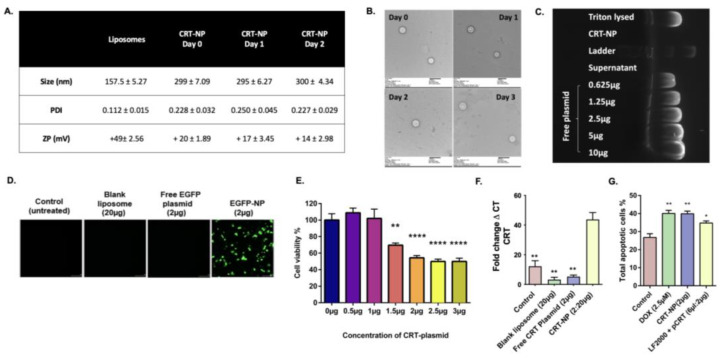
CRT-NPs showed efficient plasmid encapsulation, stability, and transfection efficiency. (**A**) CRT-NPs showed an increased hydrodynamic diameter and a reduced zeta potential compared to blank liposomes, with excellent stability in the physiological buffer for 3 days. (**B**) TEM images (scale bar—500 nm) of CRT-NPs stored in physiological buffer and obtained over four consecutive days demonstrated excellent morphological stability. (**C**). Agarose gel imaging showed that the CRT plasmid (pCRT) remained bound to the NPs, as no free pCRT was observed in the unlysed CRT-NP lane. In contrast, the released plasmid from triton-lysed CRT-NPs (~2.6 µg) exhibited a band corresponding to a 2.5 µg free pCRT band. (**D**) The transfection efficiency of CRT-NPs was confirmed using EGFP as a model plasmid. Significant GFP expression (green) was observed with transfected EGFP-NP CT26 cells at 48 h compared to the untreated control, unencapsulated NP, and free EGFP plasmid groups. (**E**) CRT-NPs induced cell death in transfected CT26 cells in a dose-dependent manner. (**F**) Transfected CRT-NP cells showed significant increase in CRT gene expression when compared to other groups in qRT-PCR. (**G**) CRT-NP-induced apoptosis was significant different compared to untreated control and comparable to positive controls such as DOX and LF™2000 + pCRT-treated cells. The statistical significance was indicated as * *p* < 0.05, ** *p* < 0.005, **** *p* < 0.0001.

**Figure 2 pharmaceutics-15-01693-f002:**
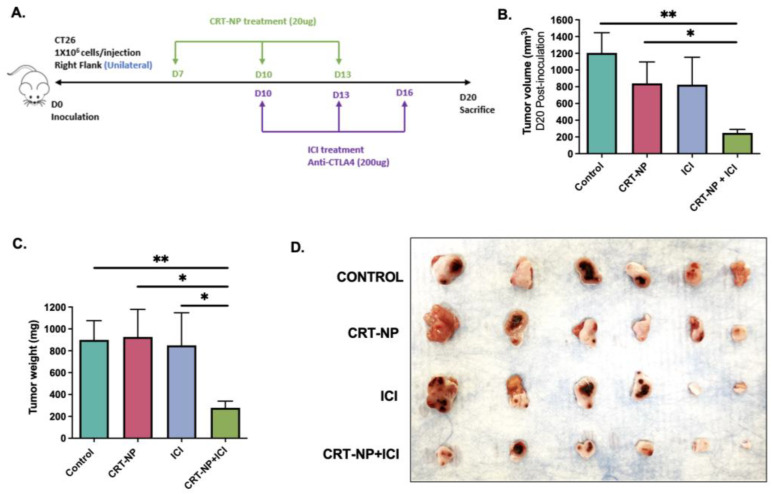
CRT-NP and antiCTLA-4 treatment efficacy in the murine model of CT26 colon cancer. (**A**) Schematics of the in vivo study design in the murine model of CT26 colon cancer. On Day 0, 1 × 10^6^ CT26 cells/mouse were injected subcutaneously in the flank region. Starting on day 7 post-inoculation, three intratumoral injections of CRT-NPs (20 µg DNA/injection) were administered in 3-day intervals. Anti-CTLA-4 immune checkpoint inhibitors (ICI) were administered 72 h after each CRT-NP injection. On Day 20, mice were sacrificed, and tumor and blood samples were collected for immune and cytokine analysis. (**B**) CRT-NP and anti-CTLA-4 treatment achieved a reduction of approximately 20–30% in tumor volumes during the treatment period. In contrast, the combination of CRT-NPs and anti-CTLA-4 significantly suppressed tumor growth rates by over 70% compared to untreated mice. (**C**) Tumor weights at the time of sacrifice showed a significant reduction in the overall weight with the treatments compared to the control. * *p* < 0.05, ** *p* < 0.005. (**D**) Images of the harvested tumors according to the treatment groups and size. On day 20 post-tumor inoculation, the combination therapy of ICI and CRT-NPs induced a significant decrease in volume by visual comparison.

**Figure 3 pharmaceutics-15-01693-f003:**
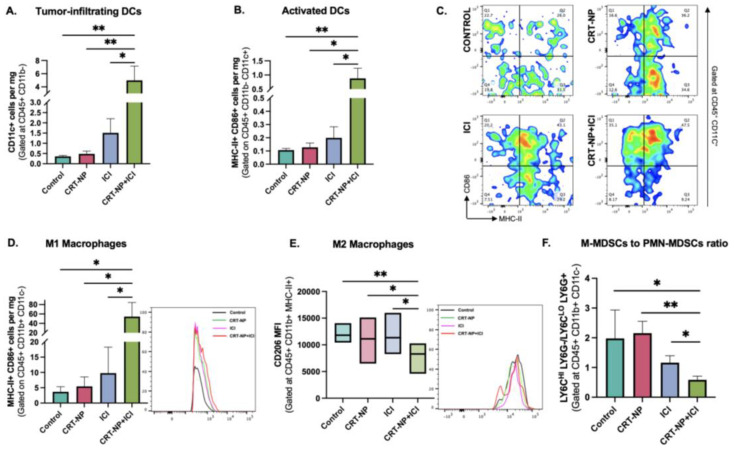
CRT-NPs synergized with anti-CTLA-4 ICI to enhance APC activation. A significant increase in (**A**) dendritic cell (CD45+ CD11c+) infiltration and (**B**,**C**) their activation (CD11c+ MHC-II+ CD86+) was observed in CRT-NP + ICI-treated tumors compared to monotherapies and control. Pro-inflammatory TME was observed with combination treatment with a significant decline in (**F**) M-MDSC (CD11b+ LY6C^Hi^ LY6G–) populations, (**E**) M2 macrophages (CD11b+ MHC-II+ CD206+), and (**D**) enhanced M1 macrophages (CD11b+ MHC-II+ CD86+) compared to other treatment groups. * *p* < 0.05, ** *p* < 0.005.

**Figure 4 pharmaceutics-15-01693-f004:**
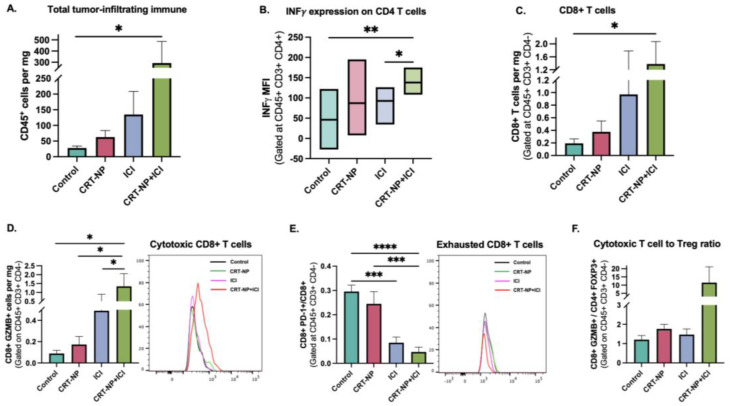
CRT-NPs and anti-CTLA-4 enhanced the T-cell immune activations in CT26 tumors. The number of (**A**) CD45+ cells, (**B**) activated CD4+ T_H_ cells (CD4+ INFγ+), (**C**) frequencies of CD8+ T cells, and (**D**) cytotoxic CD8 T cells (CD8+ Granzyme B+) was significantly higher in tumors treated with combination treatment following CRT-NP-induced re-sensitization. (**E**) Histogram of representative CD8+ T cells shows a significant decrease in PD-1+ CD8+ T cells for the CRT-NP+anti CTLA4 group relative to other groups. (**F**) A notable increase in the ratio of cytotoxic T cells and regulatory T cells (CD4+ FOXP3+) was observed in the combination treatment group compared to monotherapies. * *p* < 0.05, ** *p* < 0.005, *** *p* < 0.005, **** *p* < 0.0001.

## Data Availability

The data supporting the findings of this study are available within the article and from the corresponding author upon request.

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
