# Peer review of "Overcoming Resistance to Immune Checkpoint Inhibitor Therapy Using Calreticulin-Inducing Nanoparticle"

_pharmaceutics, 2023, doi:10.3390/pharmaceutics15061693_

Round 1

Reviewer 1 Report

In this study, the Authors aim to further extend their previous analysis of the role of in situ vaccine (ISV) administration of calreticulin-expressing nanoparticles (CRT-NPs) to induce immunogenic cell death (ICD) and antitumor immunity.

In particular, using a mouse model of syngeneic colon cancer (CT26), the Authors explore the combinatorial effect of CRT-NPs and ICI. They show that the combination of CRT-NP and anti-CTLA-4 treatments has a synergistic effect on tumor growth inhibition compared to monotherapies. Furthermore, this combinatorial treatment is able to modify the immunosuppressive microenvironment and possibly to overcome the immunoresistance to ICB.

The paper is of great interest in the field of immunotherapy with immune checkpoint inhibitors, considering the urgent need to develop new approaches capable of improving the efficacy of ICI therapy, particularly in immune-resistant tumors. However, the paper's conclusions are not fully supported by the data presented.

Specific comments

·        Do the authors have also tested  the combinatorial effect between CRT-NP and antiPD-1 therapy? The authors should at least comment  their chose to analyze the combinatorial effect of CRT-NP/ISV and anti-CTLA4 in their murine model.

·        Is the reduction in tumor growth shown in Figure 2B-D, associated with different levels of CRT expression? A comparative assessment of CRT expression in tumor sections under different experimental conditions should be shown.

·        An assessment of in vivo immunogenic cell death would allow a better characterized of the ICD-mediated effect of CRT-NP-ISV administration.

·        Graph Overlays and legends in figure 3E/F and figure 4D/E must be improved

·        As also mentioned by the authors, the presence of high levels of Treg in TME represents a crucial mechanism of tumor immune evasion also in the CT-26 colon cancer cell model. Although the authors state in the abstract session that CRT-NP/ISV and anti-CTLA4 combination therapy reduces the CD4+ Foxp3 cell population, this data was not shown. It is unclear why the authors did not include the analysis of Treg population in Figures 3 and 4.

Author Response

  1. Do the authors have also tested the combinatorial effect between CRT-NP and antiPD-1 therapy? The authors should at least comment their chose to analyze the combinatorial effect of CRT-NP/ISV and anti-CTLA4 in their murine model.
  • We did not test combined CRT-NP and anti-PD-1 therapy in the CT26 colon cancer model. Although previous research has investigated combining Anti-CTLA4 with anti-PD1 in the CT-26 model to target multiple checkpoint pathways 1,2, this project's main objective was to investigate the role of immunogenic cell death (ICD)-based modulation by CRT-NP on immunoresistance pathways and whether comparable results could be achieved when combined with anti-CTLA4. Our findings suggest that ICD agents like CRT-NP have the potential to enhance the reversal of immunoresistance when used in conjunction with immune checkpoint inhibitors like anti-CTLA4. In our future trials, we intend to explore similar combinations with anti-PD1 and anti-PDL1 to demonstrate their feasibility and efficacy.
  1. Is the reduction in tumor growth shown in Figure 2B-D, associated with different levels of CRT expression? A comparative assessment of CRT expression in tumor sections under different experimental conditions should be shown.
  • In a previous publication, we optimized the treatment dose to 2µg of CRT plasmid per 100-300mm3 of B16F10 tumor 3. As shown in Fig. 1E, the ICD effects plateaued beyond a certain CRT-plasmid dose (approximately 2µg in vitro). Therefore, we decided to employ this dosing strategy for the CT26 model.
  • It is important to emphasize that the primary aim of CRT-NP is to modulate ICD pathways. As indicated in Fig. 1F-G, the increased CRT expression correlates with apoptosis, similar to the positive control Doxorubicin. Based on the current and past characterization of CRT expression in the B16F10 and CT26 models, we propose that the optimal immune priming with CRT-NP is sufficient to significantly enhance efficacy in combination with anti-CTLA4. Future clinical trials in human settings will explore whether the principles of allometry can be successfully applied and scaled to a patient setting using this particular nanoparticle.
  1. An assessment of in vivo immunogenic cell death would allow a better characterized of the ICD-mediated effect of CRT-NP-ISV administration.
  • We have already explored the concept of ISV based ICD in published studies using CRT-NP to deliver a CRT expression plasmid for its in vivo generation in melanoma cell for tumor immune priming4 (Fig.1). This formed the basis of this paper’s anti-CTLA4 studies.
  1. Graph Overlays and legends in figure 3E/F and figure 4D/E must be improved
  • As suggested, we updated Fig. 3E/F and Fig. 4D/E.
  1. As also mentioned by the authors, the presence of high levels of Treg in TME represents a crucial mechanism of tumor immune evasion also in the CT-26 colon cancer cell model. Although the authors state in the abstract session that CRT-NP/ISV and anti-CTLA4 combination therapy reduces the CD4+ Foxp3 cell population, this data was not shown. It is unclear why the authors did not include the analysis of Treg population in Figures 3 and 4.
  • CRT-NP ISV indeed demonstrated a trend towards reduction in the Treg population (CD4+ FOXP3+), but the p-value of the comparison between the control and CRT-NP+ICI groups was 0.09. These findings have now been included in Fig. 4F and discussed in detail in the Results section (3.5).

Literature cited

1          Fiegle, E. et al. Dual CTLA-4 and PD-L1 Blockade Inhibits Tumor Growth and Liver Metastasis in a Highly Aggressive Orthotopic Mouse Model of Colon Cancer12. Neoplasia 21, 932-944, doi:10.1016/j.neo.2019.07.006 (2019).

2          Rupp, T. et al. Anti-CTLA-4 and anti-PD-1 immunotherapies repress tumor progression in preclinical breast and colon model with independent regulatory T cells response. Transl Oncol 20, 101405, doi:10.1016/j.tranon.2022.101405 (2022).

3          Sethuraman, S. N. et al. Novel calreticulin-nanoparticle in combination with focused ultrasound induces immunogenic cell death in melanoma to enhance antitumor immunity. Theranostics 10, 3397-3412, doi:10.7150/thno.42243 (2020).

4          Sethuraman, S. N. et al. Novel calreticulin-nanoparticle in combination with focused ultrasound induces immunogenic cell death in melanoma to enhance antitumor immunity. Theranostics 10, 3397-3412, doi:10.7150/thno.42243 (2020).

5          Wang, W., Li, W., Ma, N. & Steinhoff, G. Non-viral gene delivery methods. Current pharmaceutical biotechnology 14, 46-60 (2013).

6          Daraee, H., Etemadi, A., Kouhi, M., Alimirzalu, S. & Akbarzadeh, A. Application of liposomes in medicine and drug delivery. Artif Cells Nanomed Biotechnol 44, 381-391, doi:10.3109/21691401.2014.953633 (2016).

7          Liu, P., Chen, G. & Zhang, J. A Review of Liposomes as a Drug Delivery System: Current Status of Approved Products, Regulatory Environments, and Future Perspectives. Molecules 27, doi:10.3390/molecules27041372 (2022).

Reviewer 2 Report

The work "Overcoming resistance to immune-checkpoint therapy using calreticulin-inducing nanoparticle" is interesting and promising for the publication. However, some minor modifications are needed.

1. A calreticulin-inducing nanoparticle work was published with the same preparation method. Please mention and explain the difference in the manuscript. Moreover, even preparing by the same method, the particle properties are different. Please explain.

2. The characteristics of particles should be evaluated more such as surface properties, the stability of plasmid corresponding to the physical stability.

3. The change of CRT expression on cells (in vitro) are required to evaluated.

4. The change of CRT and CTLA-4 expression on tumor is also required to report

Author Response

  1. A calreticulin-inducing nanoparticle work was published with the same preparation method. Please mention and explain the difference in the manuscript. Moreover, even preparing by the same method, the particle properties are different. Please explain.
  • This study cloned the CRT gene into an antimicrobial resistance (AMR) gene-free NTC plasmid backbone (Nature Technology Corp., NE), an ideal non-viral vector for translation to human trials vs. pCMV3 expression systems that we published previously with AMR genes, which would hindered the licensing of our CRT-NP product 3. CRT-NP synthesized using NTC-plasmids met the quality control criterion (200-300nm size, and positive zeta potential)5 needed for inducing rapid, efficient and robust gene expression in cells.
  1. The characteristics of particles should be evaluated more such as surface properties, the stability of plasmid corresponding to the physical stability.
  • As suggested, we further characterized the CRT-NP stored at 4°C over a period of 3 days using SEM imaging. We observed no significant changes in NP shape and morphology (Fig. 1B). Consistent with the SEM physical stability results, EGFP-NP stored at 4°C for up to 3 days didn’t shown any significant alteration in the transfection capacity of the NP. These new results were included in the sections 3.1 and 3.2.
  1. The change of CRT expression on cells (in vitro) are required to evaluated.
  • The revised include data on changes of CRT-expression estimated using RT-PCR in-vitro (Fig. 1F). These findings were incorporated into the modified Results section 3.2.
  1. The change of CRT and CTLA-4 expression on tumor is also required to report.
  • Analysis of the expression of CTLA4 on CD45+ immune cells using flow cytometry showed an increasing trend with the combination therapy but was not statistically significantly different between the various treatment groups (Supplementary Fig. 1B). This outcome is not surprising, considering that checkpoint expressions were solely evaluated in the tumor-derived cells at the end of the experiment, while these events are immunologically highly dynamic and can change in various other organs that were not evaluated (blood/lymph nodes). To gain further insights into the associated mechanisms, in future studies, we will concentrate on assessing T-cell expression of CTLA1 and antiPD1 in other tissues (such as blood, lymph nodes, and spleen) and at multiple time points following CRT-NP treatment.

Literature cited

1          Fiegle, E. et al. Dual CTLA-4 and PD-L1 Blockade Inhibits Tumor Growth and Liver Metastasis in a Highly Aggressive Orthotopic Mouse Model of Colon Cancer12. Neoplasia 21, 932-944, doi:10.1016/j.neo.2019.07.006 (2019).

2          Rupp, T. et al. Anti-CTLA-4 and anti-PD-1 immunotherapies repress tumor progression in preclinical breast and colon model with independent regulatory T cells response. Transl Oncol 20, 101405, doi:10.1016/j.tranon.2022.101405 (2022).

3          Sethuraman, S. N. et al. Novel calreticulin-nanoparticle in combination with focused ultrasound induces immunogenic cell death in melanoma to enhance antitumor immunity. Theranostics 10, 3397-3412, doi:10.7150/thno.42243 (2020).

4          Sethuraman, S. N. et al. Novel calreticulin-nanoparticle in combination with focused ultrasound induces immunogenic cell death in melanoma to enhance antitumor immunity. Theranostics 10, 3397-3412, doi:10.7150/thno.42243 (2020).

5          Wang, W., Li, W., Ma, N. & Steinhoff, G. Non-viral gene delivery methods. Current pharmaceutical biotechnology 14, 46-60 (2013).

6          Daraee, H., Etemadi, A., Kouhi, M., Alimirzalu, S. & Akbarzadeh, A. Application of liposomes in medicine and drug delivery. Artif Cells Nanomed Biotechnol 44, 381-391, doi:10.3109/21691401.2014.953633 (2016).

7          Liu, P., Chen, G. & Zhang, J. A Review of Liposomes as a Drug Delivery System: Current Status of Approved Products, Regulatory Environments, and Future Perspectives. Molecules 27, doi:10.3390/molecules27041372 (2022).

Reviewer 3 Report

The study proposed addresses one of the main limitation of cancer therapy, which is the contribution of the immunosuppressive TME in impairing even the most advanced immunotherapeutic treatments. 

The data of the combination therapy of ICI and CRT-NP are interesting and I would dedicate more space in the results description (maybe shortening materials and methods). The mechanistic significance of the combination could be better described in the results session.

Minor comments:

- line 214 : probably a mistake, please check and correct.

- please describe somewhere in the manuscript the rationale of choosing liposome-based NP over all the other NP types.

In general, the manuscript is easily understandable, only minor typo check required.

Author Response

  1. line 214 : probably a mistake, please check and correct.
  • We corrected line 214 to now read “Figures and Tables”.
  1. please describe somewhere in the manuscript the rationale for choosing liposome-based NP over all the other NP types
  • We employed DOTAP and Cholesterol lipids based cationic since they are biodegradable, and can attain efficient plasmid loading and transfection 6,7. This was added in the discussion section (line# 337-339)

Literature cited

1          Fiegle, E. et al. Dual CTLA-4 and PD-L1 Blockade Inhibits Tumor Growth and Liver Metastasis in a Highly Aggressive Orthotopic Mouse Model of Colon Cancer12. Neoplasia 21, 932-944, doi:10.1016/j.neo.2019.07.006 (2019).

2          Rupp, T. et al. Anti-CTLA-4 and anti-PD-1 immunotherapies repress tumor progression in preclinical breast and colon model with independent regulatory T cells response. Transl Oncol 20, 101405, doi:10.1016/j.tranon.2022.101405 (2022).

3          Sethuraman, S. N. et al. Novel calreticulin-nanoparticle in combination with focused ultrasound induces immunogenic cell death in melanoma to enhance antitumor immunity. Theranostics 10, 3397-3412, doi:10.7150/thno.42243 (2020).

4          Sethuraman, S. N. et al. Novel calreticulin-nanoparticle in combination with focused ultrasound induces immunogenic cell death in melanoma to enhance antitumor immunity. Theranostics 10, 3397-3412, doi:10.7150/thno.42243 (2020).

5          Wang, W., Li, W., Ma, N. & Steinhoff, G. Non-viral gene delivery methods. Current pharmaceutical biotechnology 14, 46-60 (2013).

6          Daraee, H., Etemadi, A., Kouhi, M., Alimirzalu, S. & Akbarzadeh, A. Application of liposomes in medicine and drug delivery. Artif Cells Nanomed Biotechnol 44, 381-391, doi:10.3109/21691401.2014.953633 (2016).

7          Liu, P., Chen, G. & Zhang, J. A Review of Liposomes as a Drug Delivery System: Current Status of Approved Products, Regulatory Environments, and Future Perspectives. Molecules 27, doi:10.3390/molecules27041372 (2022).

Round 2

Reviewer 1 Report

This Reviewer considers each of the requests made, fully provided by the Authors and thus considers the new version of the manuscript acceptable for publication in Pharmaceutics